# Deep reinforcement learning for optimal experimental design in biology

**Neythen J. Treloar**[1]*, **Nathan Braniff**[2], **Brian Ingalls**[2], **Chris P. Barnes**[1,3]*

**1** Department of Cell and Developmental Biology, University College London, London, United Kingdom, **2** Department of Applied Mathematics, University of Waterloo, Waterloo, Canada, **3** UCL Genetics Institute, University College London, London, United Kingdom

* neythen.treloar.14@ucl.ac.uk (NJT); christopher.barnes@ucl.ac.uk (CPB)

## Abstract

The field of optimal experimental design uses mathematical techniques to determine experiments that are maximally informative from a given experimental setup. Here we apply a technique from artificial intelligence—reinforcement learning—to the optimal experimental design task of maximizing confidence in estimates of model parameter values. We show that a reinforcement learning approach performs favourably in comparison with a one-step ahead optimisation algorithm and a model predictive controller for the inference of bacterial growth parameters in a simulated chemostat. Further, we demonstrate the ability of reinforcement learning to train over a distribution of parameters, indicating that this approach is robust to parametric uncertainty.

**Data Availability Statement:** The code and examples are available in the RED package available on GitHub: https://github.com/ucl-cssb/RED. Data to reproduce the main plots in this manuscript are available on Zenodo: https://

## Author summary

Biological systems are often complex and typically exhibit non-linear behaviour, making accurate model parametrisation difficult. Optimal experimental design tools help address this problem by identifying experiments that are predicted to provide maximally informative data for parameter inference. In this work we use reinforcement learning, an artificial intelligence method, to determine such experiments. Our simulation studies show that this approach allows uncertainty in model parameterisation to be directly incorporated into the search for optimal experiments, opening a practical avenue for training an experimental controller. We present this method as complementary to existing optimisation approaches and we anticipate that artificial intelligence has a valuable role to play in the future of optimal experimental design.

## Introduction

A key goal of systems and synthetic biology is to apply engineering principles in service of understanding and building biological systems. Such approaches rely on inference of mechanistic models and downstream model-based prediction of cellular systems. Biological systems are typically complex, highly non-linear, and noisy, making development of accurate models a challenging task [1–3]. Furthermore, characterisation experiments can be resource-intensive

zenodo.org/record/6521194#.Y2D-HVLML9E. All other relevant data are within the manuscript and its Supporting information files.

**Funding:** NJT and CPB received funding from the European Research Council (ERC) under the European Union's Horizon 2020 research and innovation programme (Grant No. 770835). BI and NB were supported by a Discovery Grant from Canada's Natural Sciences and Engineering Research Council (NSERC). The funders had no role in study design, data collection and analysis, decision to publish, or preparation of the manuscript.

**Competing interests:** The authors have declared that no competing interests exist.

and time consuming; efficient use of experimental effort is critical. The field of optimal experimental design (OED) uses mathematical techniques to identify experiments that will provide maximally informative characterisation data within the constraints of a fixed experimental capacity. OED tools are often used to establish accurate model-based control methods; there are close ties between the development of OED and control theory [4–6].

Application of OED to nonlinear biological systems is challenging; techniques for efficient OED in this context have been developed via Bayesian methods [7–11] and methods based on optimisation of the Fisher information [12–18]. Here, we use reinforcement learning (RL), a branch of machine learning, to develop a novel Fisher information-based OED method for parametrisation of non-linear models of biological systems. Although we focus on an application relevant to synthetic biology, the method is general and can be applied to any OED model parametrisation task.

Reinforcement learning methods learn control policies from data that is generated either by simulation or by interaction with a real system. Much of reinforcement learning research has focused on games [19–21], but its effectiveness has also been shown in optimising chemical reactions [22], controlling plasmas for nuclear fusion [23] and computer chip design [24]. Standard OED methods determine experiments by optimising with respect to a model of the system. This can require complex non-linear programming or integration over posterior distributions to calculate online experimental inputs. In contrast, a reinforcement learning agent chooses its actions based on its experience, which is generated through trial and error interaction during a training process. Although a satisfactory training process can be rather lengthy, the resulting trained agent can act as an online experimental controller, rapidly identifying optimal experimental inputs.

Model-based optimal experimental design faces a fundamental challenge: the optimization problem is formulated in terms of the underlying model, whose accuracy can only be guaranteed if the parameter values to be inferred are already known. Of course, if those parameter values are known then there's no need to design experiments to infer them. An iterative OED estimation approach can be employed to resolve this problem [16–18]: begin with an initial guess for the parameter values, then iteratively apply OED, using the resulting experiments to improve the parameter estimates. Even in this case, the performance of the resulting experimental design will still be dependent on the initial guess of parameter values. Bayesian approaches to OED incorporate robustness to this initial data by beginning with a probability distribution of parameter estimates (a *prior*) rather than a single guess [25]. However, Bayesian approaches typically involve computationally expensive integration, which can make them unsuitable for real time online experimental design, and are restricted to specific distribution types [7, 9]. Machine learning methods have the potential to alleviate some of these problems. For example, Deep Adaptive Design [26] was developed to reduce the computational cost of Bayesian approaches. This method was shown to be effective for real time experimental design, but has limitations such as reliance on a differentiable objective function and limited ability to explore design space. Reinforcement learning has been proposed as a method for experimental design that avoids these limitations [27]. Here, we propose a reinforcement learning approach to tackle the problem of limited parameter knowledge while also being capable of rapid online experimental design for time course experiments.

In the presentation below, we develop our novel reinforcement learning OED algorithm by iterative improvement of a baseline Fitted Q-learning (FQ-learning) approach. We begin by outlining our proposed formulation of reinforcement learning for OED and introducing a model of bacterial growth in a chemostat, which will be used as an application. We then apply the FQ-learning algorithm [28], which we have previously shown to be effective in controlling biological systems [29], to the chemostat growth model. We use this initial analysis, in which

we unrealistically assume prior knowledge of the true parameters, to explore the baseline performance of the RL-OED approach. Next, we consider an alternative formulation of the learning agent, based on a recurrent neural network. We demonstrate, using a direct measure of value-function construction, that the corresponding algorithm has the capacity to effectively predict the optimality of experiments independently of prior parameter estimates. However, an assessment of OED performance reveals that this strategy yields unsatisfactory performance. To address that issue, we again alter the learning agent formulation, this time extending its behaviour to a continuous action space, through application of the Twin Delayed Deep Deterministic policy gradient (T3D) algorithm [30]. We call the resulting RL approach the Recurrent T3D (RT3D) algorithm. Successful OED performance of this algorithm is demonstrated. Finally, we demonstrate that the RT3D algorithm is robust to uncertainty by training the agent over an ensemble of model instances drawn from a parameter distribution, resulting in a robust controller and illustrating the algorithm's ability to design experiments with limited prior knowledge. Throughout, we compare the OED performance of the reinforcement learning controllers with both a one step ahead optimiser (OSAO) and a model predictive controller (MPC).

## Results

### Reinforcement learning for optimal experimental design

We focus our work on Fisher information-based experimental design. D-optimal design aims to maximise the determinant of the Fisher information matrix (FIM). For linear models with Gaussian measurement error, this goal is equivalent to minimising the volume of the confidence ellipsoid of the resulting parameter estimates [31]. This approach has been demonstrated to be useful even for non-linear systems [12–18]. Fig 1A shows the expected outcome from a hypothetical OED application. Input profiles for two imagined experiments are shown, representing equivalent experimental effort. Data from the corresponding system outputs are used to infer model parameter values, resulting in expected confidence ellipsoids of parameter estimates. This comparison indicates that a poorly designed experiment (left panel: large confidence ellipsoid and low D-optimality, defined as the logarithm of the determinant of the Fisher information matrix) is less informative than the well designed experiment (right panel: small confidence ellipsoid and high D-optimality).

Reinforcement learning is a branch of machine learning concerned with optimising an agent's behaviour within an environment. The agent responds to observations of its environment by selecting from a set of actions that, in turn, impact the environment. From a reward structure imposed on this interaction, the agent learns an optimal behaviour *policy* (Fig 1B). The training of a reinforcement learning agent is typically implemented as a collection of *episodes*, each of which is a temporal sequence of observations and actions, repeated until a terminal state is reached. At each discrete time step, the agent receives a scalar reward that can depend on both observation and the selected action. The cumulative reward is called the *return*. The agent's goal is to learn a behaviour policy that maximises the return. In our application the return is the D-optimality score of a given experiment (see Methods for details). Many reinforcement learning algorithms construct internal estimates of the expected future return as part of their learning strategy. A *value function*, which maps each observation-action pair to an expected future return, can be learned from experience. In the work presented here, we represent the value function as a neural network. A trained agent uses this value function to make decisions, choosing actions that maximise the value.

A key consideration in reinforcement learning is the exploration-exploitation trade-off: a strategy must be adopted to determine when to invest in exploration of new behaviours vs.

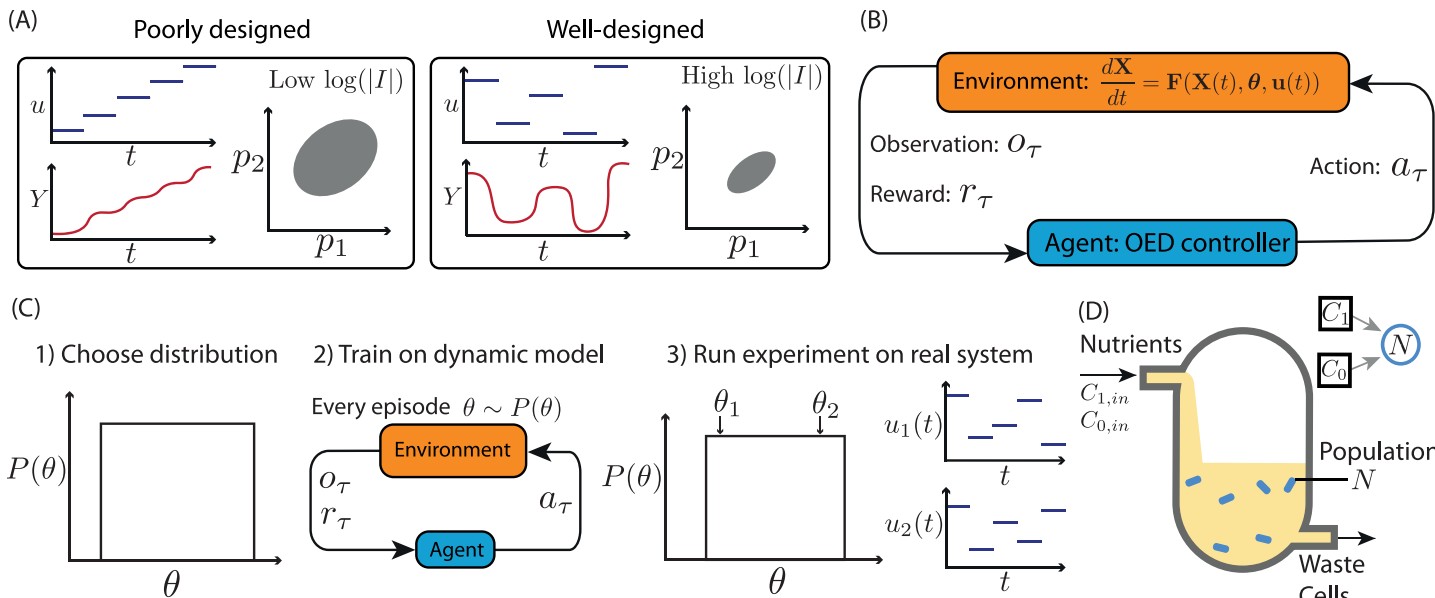

**Fig 1. Reinforcement learning for optimal experimental design.** A) A hypothetical example of a poorly designed experiment (left) corresponding to an increasing sequence of input values $u$ over time, with a resulting continual increase in the observable output $Y$. A corresponding confidence ellipse in $p_1$-$p_2$ parameter space is depicted. The logarithm of the determinant of the Fisher information matrix, $\log(|I|)$, is low. In contrast, a hypothetical well-designed experiment (right), which maximises the determinant of the Fisher information matrix, corresponds to non-intuitive choices of input, and a resulting dynamic response in the output. The corresponding confidence ellipse is tight and the determinant of the Fisher information is high. B) Optimal experimental design formulated as a reinforcement learning problem. The model dynamics, **F**, describe the rate of change of the state vector **X** in terms of model parameters $\boldsymbol{\theta}$ and input **u**. For each time step, $\tau$, an observation of the system, $o_\tau$, is provided to the agent which chooses an action, $a_\tau$, to apply over that time step and receives a corresponding reward, $r_\tau$. C) Training over a parameter distribution. 1) A distribution of parameters is chosen (shown as uniform). 2) The RL controller is trained. Each episode employs a model parametrisation $\theta$ sampled from the distribution. 3) When acting as a feedback controller, the trained RL agent designs near optimal experiments across the parameter distribution. For example well-designed experiments will be executed for either $\theta_1$ and $\theta_2$ (inputs $\mathbf{u}_1(t)$ and $\mathbf{u}_2(t)$ respectively). D) Model of an auxotrophic bacterial strain growing in a chemostat. The nutrient inflows, $C_{in}$ and $C_{0,in}$, can be controlled as part of an experiment.

exploiting current knowledge to adopt a 'best' policy. We employ an *epsilon-greedy* strategy for addressing this trade-off, as follows. We define the explore rate, $\epsilon$, as the probability that the agent chooses its action arbitrarily (i.e. uniformly over the range of allowed actions) as opposed to selecting the 'best' action defined by its learned policy. We define a schedule for $\epsilon$ so that it is equal to 1 at the beginning of training (ensuring wide exploration initially) and gradually decays to 0 by the end of training (allowing the trained agent to fully exploit the knowledge it has gained). For this study, we first use a continuous-observation, discrete-action reinforcement learning formulation called Neural FQ-learning [28]. We then apply a continuous-space, continuous-action algorithm called Twin Delayed Deep Deterministic Policy Gradient [30].

Rather than insisting that the sequence of experimental inputs be decided prior to the start of the experiment, which would be an open-loop design, we allow the inputs to be selected based on measurements of the system output, a closed-loop design. Reinforcement learning algorithms learn by observing discrete time series. We thus divide an experimental time series into $\mathcal{T}$ discrete time steps. Below, we use $t$ to refer to time and $\tau$ as the index of these discrete time steps: $\tau = 1, 2, \ldots, \mathcal{T}$. The constant input applied to the system during each time step is determined from an experimental measurement taken at the beginning of the time step. The time steps are thus sample-and-hold intervals. We define an **experimental design** as a sequence of experimental inputs, one for each time step in the experiment. Other experimental features such as initial conditions, environmental settings, and experiment length are

presumed fixed. In our construction the experimental inputs are determined by a reinforcement learning agent. Before experiments begin, the agent undergoes training on a dynamical model of the system of interest. In our preliminary analysis this is done assuming the true parameters are known. We later demonstrate developments that relax this assumption and thus yield a general and robust approach to OED. Training is carried out over a number of single experiment simulations, called **episodes**. During this model-based training, the reinforcement learning agent has the goal of maximising the logarithm of the determinant of the FIM over each experiment. Once training is complete, the trained agent can act as a feedback controller to provide real-time inputs to the experimental system.

To formulate an OED problem in a reinforcement learning framework, we begin by defining (i) the agent's environment, (ii) the observations available to the agent, (iii) the actions the agent can take, and (iv) the reward function that guides the agent towards optimal behaviour (Fig 1B). The agent's environment is provided by a pre-established model of the experimental system. In our case the model is governed by a set of differential equations. At the beginning of each time step, the agent is supplied with an observation, $o_\tau$. In our preliminary formulation, described below, the observation consists of the index of the current time step, $\tau$, the system output, $\mathbf{Y}_\tau$ and an estimate of the FIM, $I_\tau$. In subsequent developments of the algorithm, we use an alternative observation composed of the time step index $\tau$, system output, $\mathbf{Y}_\tau$, and the past trajectory of experimental observations and inputs. In the latter case the agent makes decisions without knowledge of the true system parameters. The action taken by the agent at each time step is the experimental input, $\mathbf{u}_\tau$, to be applied for the duration of the time step. The reward provided to the agent during training is the change in the logarithm of the determinant of the Fisher information matrix from the previous time step $r_\tau = \log |I_\tau| - \log |I_{\tau-1}|$. Consequently, over an experiment, the cumulative reward (i.e. the return) is the log of the determinant of the accumulated Fisher information matrix, and so the agent's optimisation objective is equivalent to maximising the D-optimality score (see Methods).

One of the key advantages of the reinforcement learning approach is the flexibility to incorporate different experience during training. Below, we make use of this flexibility to develop a reinforcement learning strategy that avoids dependence on accurate estimates of the parameter values. In that method, training is carried out on model simulations, over a range of parametrisations. In each episode, the agent learns to maximise the FIM-dependent reward, which depends on the parameter values used in the corresponding simulation. The trained agent can then be deployed as an experimental controller, making decisions solely based on experimental observations and assuming no specific knowledge of system parameters. Here, prior knowledge of parameter values is incorporated into the distribution of parameters over which training occurs, which can be of arbitrary shape. It should be noted that this approach is still reliant on an assumption that the model structure is accurate. This robust method of design of experiments for parameter inference is summarised in Fig 1C.

## Chemostat bacterial growth model

The genetic engineering of bacterial strains can have unintended and unpredictable impacts on their growth dynamics, caused, e.g., by increased metabolic burden or by cross talk among regulatory pathways. Consequently, determining the growth characteristics of bacterial strains can be an important task in the field of synthetic biology. A thorough investigation of growth dynamics can be carried out by implementing a range of nutrient conditions in a flow-through bioreactor. Affordable tabletop bioreactors [32–35] have become widely available, so this setup is feasible for most synthetic biology labs. Here, we consider parameter inference for a model of auxotrophic bacterial growth in a chemostat (Fig 1D), where an auxotroph is a bacterial

strain whose growth is reliant on a specific nutrient, e.g. a specific amino acid. We suppose the agent controls the concentration of carbon source, $C_{1,in}$, and auxotrophic nutrient, $C_{0,in}$, in the inflow. The bacterial population, $N$, is the only measured output. The concentrations $C_1$ and $C_0$ are hidden state variables. The system equations are:

$$\mu = \mu_{max} \frac{C_1}{K_1 + C_1} \frac{C_0}{K_0 + C_0}$$

$$\frac{d}{dt} C_0 = q(C_{0,in} - C_0) - \frac{1}{\gamma_0} \mu N$$

$$\frac{d}{dt} C_1 = q(C_{1,in} - C_1) - \frac{1}{\gamma_1} \mu N$$

$$\frac{d}{dt} N = (\mu - q)N$$

where $\mu_{max}$ is the maximal growth rate, $K_1$ is the half-maximal auxotrophic nutrient concentration, $\gamma_1$ is the yield on this nutrient, $K_0$ is the half-maximal concentration for the carbon source, with corresponding yield $\gamma_0$, and $q$ is the flow-through rate.

Here we consider experimental design with the goal of inferring parameter values for this model. We suppose that each experiment consists of a sequence of ten two-hour intervals. In each interval the experimental inputs $C_{1,in}$ and $C_{0,in}$ are assigned constant values between the minimum and maximum bounds of 0.01 and 1 g L$^{-1}$. We suppose that the initial conditions are fixed for each experiment as $N = 20 \times 10^9$ cells L$^{-1}$ and $[C_0, C_1] = [1, 0]$ g L$^{-1}$. Model simulations were performed with parameter values as in Table 1. In previous work we found that the parameters $\gamma_0$ and $\gamma$ are practically unidentifiable for this model implementation [36]. We thus focus on designing experiments to infer the values of parameters $K_0$, $K_1$ and $\mu_{max}$.

## Fitted Q-learning for optimal design of chemostat growth experiments

We begin our analysis of the chemostat model by testing the effectiveness of the FQ-learning algorithm [28] for designing optimal experiments. We trained 10 FQ-learning agents over 50,000 episodes, where each episode constitutes a single simulated experiment (see Methods for details), with model parameters set to their nominal values (Table 1). FQ-learning operates over a discrete action space. As feasible actions, we chose 10 discrete values equally spaced linearly between the minimum and maximum bounds of 0.01 and 1 g L$^{-1}$, respectively. Inputs to the neural networks were scaled to be between 0 and 1 (see Methods). This training took approximately 48 hours of training time on a computing cluster with 40 CPU cores and a GeForce GTX 1080 Ti. In this preliminary analysis we assume prior knowledge of the parameter values, which are used in the training simulations and calculation of the FIM to determine the agent's reward. Likewise, the one step ahead optimiser (OSAO) and model predictive

**Table 1. Parameters for the auxotroph system.** Nominal parameter values, along with minimum and maximum bounds, used for simulations of a bacterial culture.

| Parameter | Value | Minimum | Maximum | Unit | Source |
|---|---|---|---|---|---|
| q | 0.5 | | | h$^{-1}$ | Experimentally controllable |
| $\gamma_0$ | $4.8 \times 10^{11}$ | | | cells g$^{-1}$ | [37] |
| $\gamma_1$ | $5.2 \times 10^{11}$ | | | cells g$^{-1}$ | [38] |
| $\mu_{max}$ | 1 | 0.5 | 2 | h$^{-1}$ | [39] |
| $K_0$ | $6.8 \times 10^{-5}$ | $10^{-5}$ | $10^{-4}$ | g L$^{-1}$ | [38] |
| $K_1$ | $4.9 \times 10^{-4}$ | $10^{-4}$ | $10^{-3}$ | g L$^{-1}$ | [38] |

controller (MPC), which are implemented for comparison, optimise with respect to the true parameters (see Methods for details).

Ten FQ-agents were trained. The experimental input profiles and resulting system trajectories for a human-chosen rational design, OSAO, MPC, and the best performing FQ-agent are shown in Fig 2A–2D. Fig 2E shows the training performance of the ten FQ-agent instances and the equivalent performance (far right) of the optimisers and rational design. The FQ-agents successfully learn throughout the training process, as shown by the increase in optimality score as training progresses, with an average final optimality score of 16.10. The best FQ-agent performs significantly better than the OSAO, but not as well as the MPC. As shown, there is significant variance in the performance of the FQ-agents.

To assess the quality of the experimental designs, we compared their performance in generating parameter estimates, as follows. We simulated the model driven by each experimental design and added noise to the corresponding observation outputs. We then determined parameter fits to this simulated data (see Methods for details). For each method, we calculated the normalised mean squared error (MSE) and the logarithm of the determinant of the covariance matrix of independent parameter estimates. The results are shown in Table 2. The best performing FQ-learning experiment (from Fig 2E) was used for this comparison. These results show that a high optimality score is a good predictor of both a low determinant of the covariance matrix and a low error in the inferred parameter values. As expected from the D-optimality scores, the MPC has greatly outperformed the rational design and the OSAO, while the

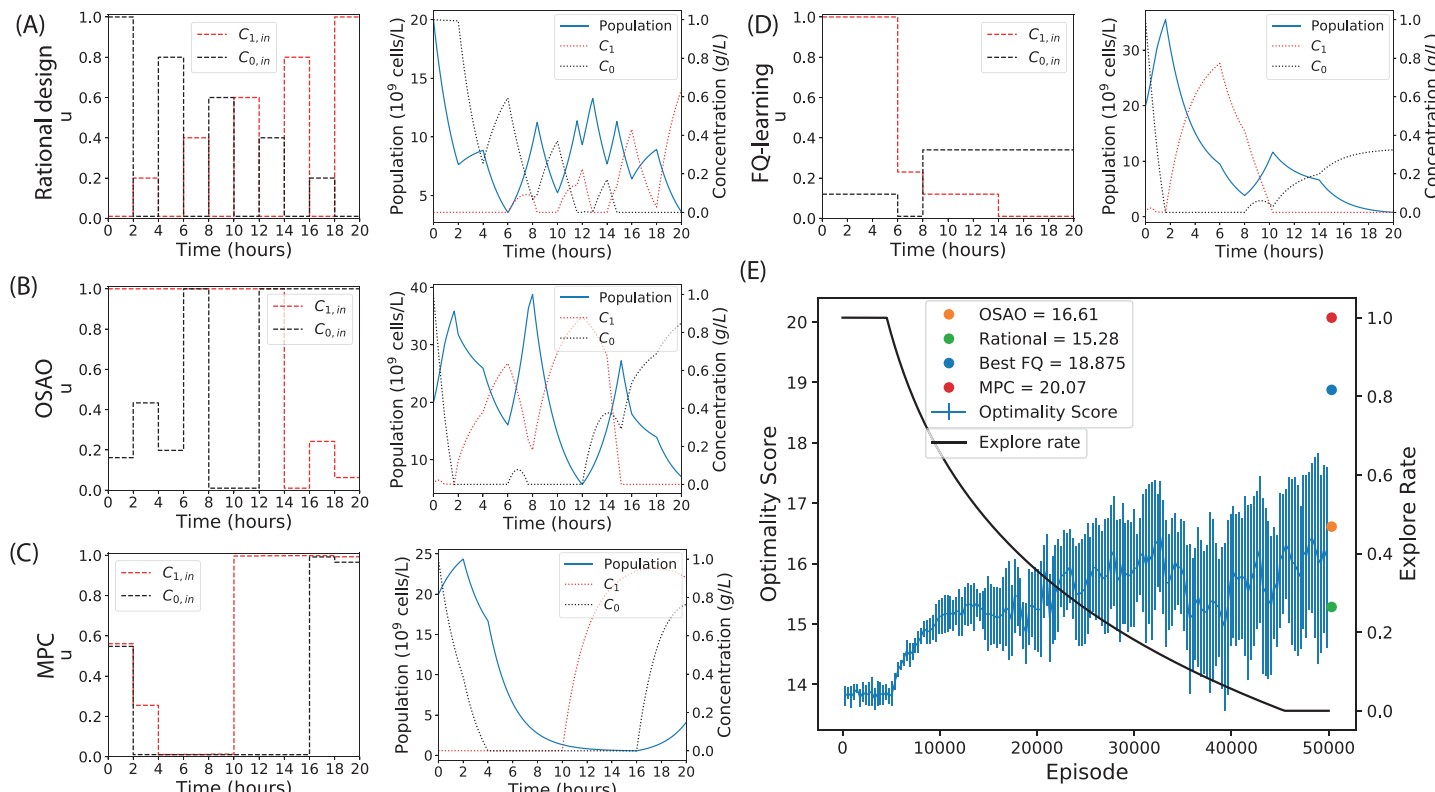

**Fig 2. Optimal experimental design to infer the values of model parameters for an auxotrophic bacterial strain growing in a chemostat.** Control inputs chosen by (A) rational design, (B) one step ahead optimisation, (C) model predictive control, and (D) FQ-learning, along with the corresponding system trajectories. (E) Training progress of ten independent FQ-agents over 50000 episodes, their shared explore rate, and (far-right) the scores of the MPC, OSAO and rational design. The mean FQ-agent return is shown, along with error bars indicating one standard deviation.

**Table 2. Comparison of the performance of optimal experimental designs in supporting parameter inference for the auxotroph model.**

|  | Rational | OSAO | FQ | MPC |
|---|---|---|---|---|
| D-optimality | 15.28 | 16.6 | 18.88 | 20.07 |
| Normalised MSE | 0.37 | 0.30 | 0.09 | 0.068 |
| $\log|cov(\theta)|$ | -3.76 | -5.47 | -8.56 | -8.72 |

performance of the best FQ-agent is between these extremes. The results in this section and our preliminary investigations (Fig A in S1 Text, [36]) show that reinforcement learning has potential for optimal experimental design. Below, we build on this potential by developing improvements to the method.

## An agent without access to *a priori* parameter estimates can learn from observations of past time series

In the following sections we show that a reinforcement learning agent can accurately predict the D-optimality score of experiments without *a priori* knowledge of the true parameter values. As an initial assessment of such an agent, we consider six different formulations of the agent's observation and assess each variant's ability to learn a value function. Reinforcement learning is based on using experience to estimate a value function that maps observation-action pairs to the expected return obtained by taking a given action after receiving a given observation. A good value function will show low error in predicting returns. In our case the return of a full experiment is equivalent to the D-optimality score. Here, all agents choose from the same set of actions and are rewarded in the same way as in the previous section; they differ in the observations that are available to them.

- Agent Ia observes the time step index, the system measurements, and the elements of the FIM (identical to the FQ-controller in the previous section). Agent Ib is identical but does not observe the time step index.

- Agent IIa observes the current time step index and the system measurements. Agent IIb is identical but does not observe the time step index.

- Agent IIIa observes the current time step index, the system measurements, and the history of measurements and actions from the beginning of the experiment. Agent IIIb is identical but does not observe the time step index.

Here, formulation I, which makes use of *a priori* knowledge of the parameter values (via the FIM) acts as a positive control, while formulation II is a negative control which is not expected to have sufficient information to be successful. In each case the comparison between variants (a and b) reveals whether the agents learn to use the dynamics of the system, rather than simply mapping a time step to a predicted return. The return is time dependent (as demonstrated in Fig BB in S1 Text), because it is easier to gain further information about the system parameters during the early phases of the experiment, when little or no data has been acquired.

All agents were evaluated by testing their ability to learn the value of randomly chosen experiments undertaken on chemostat environments over a range of different parametrisations (Methods for details). Because the parameter values are different for each experiment, the agent must learn to infer where it is in parameter space to optimally predict the value of a given observation-action pair. The results, in terms of mean square testing error at the end of training, are shown in Table 3. When time information is included in the observation (variant *a*) all agents show relatively low error, with Agent IIIa outperforming Agent Ia, which does

**Table 3. Mean square error in predicted returns for six agents with different observation formulations.** The agent with access to previous observation timeseries but no *a priori* knowledge of parameter values (formulation III) are able to accurately learn value estimates using the dynamics of the system. Error values are one standard deviation over five repeats.

| | Agent I | | Agent II | | Agent III | |
|---|---|---|---|---|---|---|
| | *o* | MSE | *o* | MSE | *o* | MSE |
| Variant *a* | $\mathbf{Y}_\tau, I_\tau, \tau$ | $31 \pm 1.4$ | $\mathbf{Y}_\tau, \tau$ | $35 \pm 0.92$ | $\mathbf{Y}_{0:\tau}, \mathbf{u}_{0:\tau-1}, \tau$ | $22 \pm 1.5$ |
| Variant *b* | $\mathbf{Y}_\tau, I_\tau$ | $35 \pm 0.80$ | $\mathbf{Y}_\tau$ | $190 \pm 1.1$ | $\mathbf{Y}_{0:\tau}, \mathbf{u}_{0:\tau-1}$ | $23 \pm 1.0$ |

better than Agent IIa. Thus, the elements of the FIM observed by Agent Ia are valuable for learning, but improved performance can be obtained by instead observing the full experimental history. When time is omitted from the observations (variant *b*) Agent Ib and Agent IIIb have minor increases in error, while Agent IIb is unable to learn a reasonable value function.

The fact that Agent III's performance was not significantly reduced by the omission of the time index shows that it is able to use the dynamics of the system to make its predictions, consistent with it being able to use the dynamics to infer where it is in parameter space. Finally, in assessing training error relative to the testing error, we found that there was minimal overfitting for all agents as shown in Fig BA in S1 Text.

In summary, these results show that a reinforcement learning agent can use the observed dynamics of the system to make suitable value predictions in the absence of prior knowledge of the parameter values. Note that the reward is still dependent on the FIM, a function of the unknown parameter values. However, calculation of the reward is required only during the training period (when simulation parameters are known); it would not be required by the agent when deploying a trained agent on a real system. We next assessed Agent IIIa's OED performance using the same procedure. Much like our FQ agent, this OED comparison shows high variance between training repeats with the best repeat falling somewhere between the OSAO and MPC (Fig CA in S1 Text). This is likely due to the large number of action options: two experimental inputs each with 10 possible values yields 100 possible actions for which the value function needs to be learned, over each observation. Q-learning in large discrete action spaces is a difficult learning problem [40] and we hypothesise that this is why our RL agents are showing high variance. As a verification, we analysed a simplified variant of the learning problem for which the agents were given a choice of just 2 possible values for each experimental input, reducing the total number of action choices to 4. As expected, this resulted in much more consistent performance between training repeats (Fig CB in S1 Text). This conclusion is further is supported by previous work where we applied the same approach to a different system with a set of 12 action options and found similarly low variation in the resulting FQ-agents [36].

## Transition to continuous action space: The recurrent T3D algorithm

To improve the RL method's OED performance, we adopt a further refinement of the approach by transitioning to a continuous action space, thus avoiding the problems associated with Q-learning in large discrete action spaces.

The application of Q-learning methods in continuous action spaces is non-trivial and requires an additional neural network which represents the agent's policy. We developed a recurrent version of the continuous reinforcement learning algorithm Twin Delayed Deep Deterministic Policy Gradient (T3D) [30] (see Methods for details). We call this algorithm Recurrent T3D (RT3D). The internal structure of the agent is shown in Fig 3A. The main components of the agent are the memory, the value network, and the policy network. During each

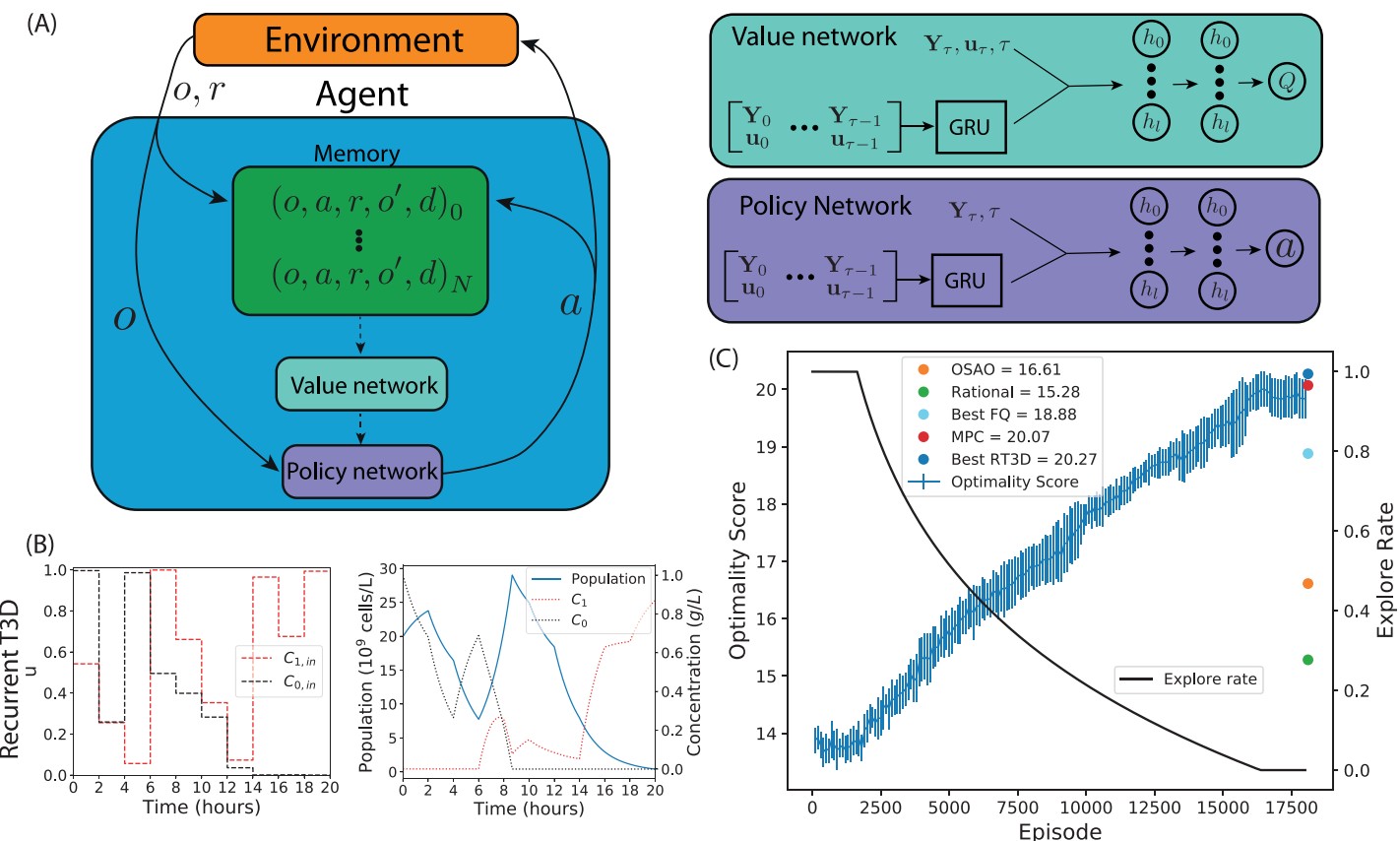

**Fig 3. The recurrent T3D algorithm.** (A) Structure of the RT3D agent (left) and the artificial neural networks used to approximate the value function (top-right) and policy (bottom-right). (B) Control inputs chosen by the best performing Recurrent T3D instance and the corresponding system trajectory. (C) Average training progress and shared explore rate of ten Recurrent T3D agents over 17500 episodes and the scores (far-right) of the MPC, OSAO and rational design. Error bars indicate one standard deviation.

time step, the observation, $o$, and reward, $r$, are determined from the environment and stored in memory. The observation also acts as input for the policy network, which chooses the action to apply. This action is also stored in memory. We define a transition as the tuple $(o, a, r, o', d)$ specifying, respectively, an observation, action, reward, a subsequent observation, and an indicator that is 1 if the episode terminated during this transition and otherwise is 0. Here $a$ is the action taken after observing $o$, while $r$ and $o'$ are the reward and observable state of the environment that result from taking action $a$. The memory contains all of the transitions observed so far during training, where $N$ is the number of transitions currently stored in memory. The agent's value and policy functions are periodically updated by training on the experience stored in the memory (dashed arrows in Fig 3A). The value function is approximated using a deep neural network. A GRU layer, a type of recurrent layer which excels at processing sequences such as time series data, takes as input all system measurements, $\mathbf{Y}$, and experimental inputs, $\mathbf{u}$, seen so far in the current experiment. The output of the GRU is concatenated with the current system measurements, $\mathbf{Y}_\tau$, the index of the current time step, $\tau$, and the current action, $\mathbf{u}_\tau$. These are fed into a feed forward network. The output of the value network is an estimate of the value of the supplied experimental input, $\mathbf{u}_\tau$, given the sequence of observables. The policy network is structured similarly, but does not take the current experimental

**Table 4. Performance metrics for optimal experimental design on the chemostat growth model.** The best performing R3TD agent is reported.

|  | Rational | OSAO | FQ | MPC | RT3D |
|---|---|---|---|---|---|
| D-optimality | 15.28 | 16.6 | 18.88 | 20.07 | 20.27 |
| Normalised MSE | 0.37 | 0.30 | 0.09 | 0.068 | 0.038 |
| $\log|cov(\theta)|$ | -3.76 | -5.47 | -7.78 | -8.72 | -11.85 |

input $\mathbf{u}_\tau$ as input. The output of the policy network is the experimental input that is estimated to have the highest value for the given observation.

## Recurrent T3D can design optimal experiments for a bacterial strain growing in a chemostat

We applied the RT3D algorithm to OED on the chemostat model. Here the RT3D agent uses the nominal parameter values during training to generate the FIM (to determine the reward), while the OSAO and MPC have access to the these parameter values for their optimisation calculations. Ten RT3D agents were trained for 17500 simulation episodes, this amounts to about 10–15 hours of computation time (see Methods). The RT3D algorithm chooses experimental inputs continuously between the maximum and minimum bounds. Inputs to the neural networks were scaled to be between 0 and 1 (see Methods). The experimental input profiles and resulting system trajectories for the best performing RT3D agent are shown in Fig 3B. Fig 3C shows the training performance of the ten RT3D agents and the equivalent performance of the OSAO, MPC, best FQ-agent and rational designs. The average RT3D optimality score was 19.84. In comparison to the FQ-controllers the average optimality score has increased to a level comparable to the MPC and the training performance is stable, with only minor differences between each of the 10 independently trained RT3D agents. The MPC reaches a slightly higher optimality score than the mean of the RT3D controllers, but the best RT3D agent performs better than the MPC. As before, we test the performance in terms of acuracy of parameter estimates, summarised in Table 4. As expected, the best performing RT3D agent has outperformed the other designs in terms of parameter covariance and parameter error. Again, we see the correlation of high D-optimality with low parameter error and low parameter covariance. From these results we can conclude that the recurrent RT3D controllers are performing at a similar level to an MPC controller and that the RT3D algorithm has significantly improved performance over the FQ algorithm with significantly lower training cost in terms of the amount of data and computation time.

## Recurrent T3D can be generate optimal experiments over a parameter distribution

Finally, we demonstrate the use of RT3D to design experiments assuming minimal knowledge of the system parameters. As shown previously, the agent is able to infer its position in parameter space using the trajectory of experimental inputs and measurements to predict the optimality score of an experiment. To assess OED performance, we trained the RT3D controller over a parameter distribution as laid out in Fig 1C. Here, each episode is initialised using a different parameter set sampled from the uniform distribution indicated in Table 1. The training performance in Fig 4A shows that RT3D controllers can successfully learn to optimise the objective over the distribution of parameters.

We carried out a comparison of the RT3D OED success, using the best performing agent from Fig 4A. We first show that this agent performs significantly better than an MPC that designs an experiment using the mean of the given parameter distribution (Table A and Fig D

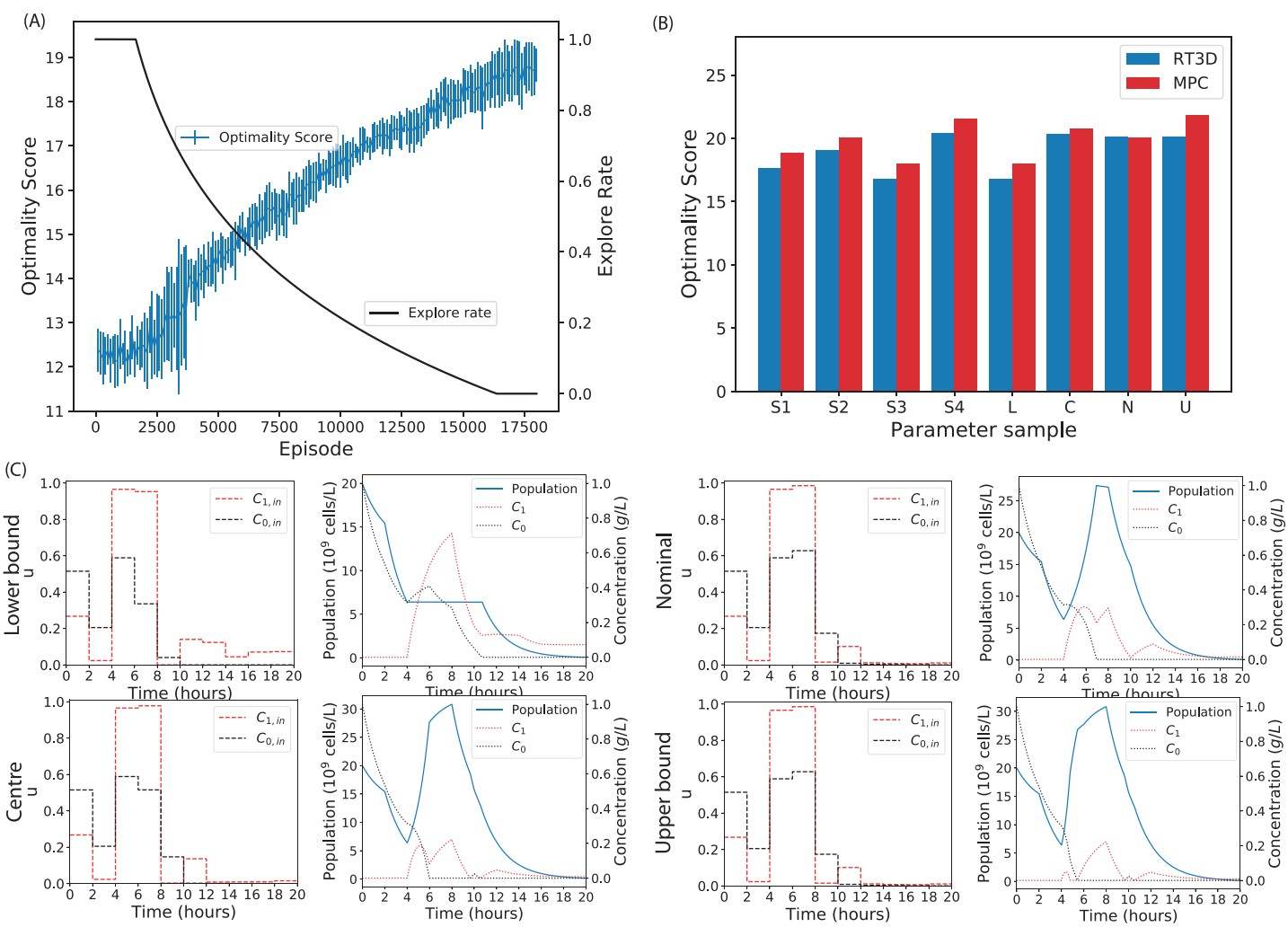

**Fig 4. The RT3D algorithm trained over a parameter distribution.** (A) Average training progress and shared explore rate of ten Recurrent T3D agents over 17500 episodes. For each episode the model parameters are sampled from a distribution. This was then averaged across the 10 repeats. Error bars indicate one standard deviation. (B) The optimality score of the best performing agent in panel A compared to an MPC with *a priori* parameter knowledge for different parameter samples. Four randomly sampled parametrisations, S1, S2, S3 and S4 are shown along with the lower and upper bounds of the parameter space, L and U, the nominal parameters from literature, N, and the centre of the distribution, C. (C) The experimental designs of the RT3D controller at different parameter samples. Here the RL controller is trained on a distribution over parameters and adapts its experimental design for different points in parameter space. For each experimental design the system is initialised with different parameter values: L, U, N and C.

in S1 Text). In this comparison, the RT3D agent has a distinct advantage because the MPC controller does not use the form of the distribution in its design. Next, we give the MPC controller an intrinsic advantage. We compared the performance, in terms of the optimality score, of the RT3D controller against an MPC controller that uses *a priori* knowledge of the sampled parameters. (Recall that the RT3D controller has only knowledge gained by training against the parameter distribution.) Eight samples were investigated. Four of these (S1, S2, S3, S4) were sampled from the distribution. The remaining four were chosen specifically to reveal the behaviour of the RT3D controller across the parameter distribution. These were (N) the nominal parameter values, (L) the lower bounds, (U) the upper bounds, and (C) the centre of the distribution. The corresponding optimality scores are shown in Fig 4B. The parameter values for each sample can be found in Table B in S1 Text. For every sample, the RT3D agent has

performed nearly as well as the MPC that has total system knowledge. From these results, we conclude that, by training over a parameter distribution, the RT3D controller can generate near optimal experiments across the whole distribution.

The experiments the RT3D agent designed at the L, U, C and N parameter samples and resulting system trajectories are plotted in Fig 4C. The experimental inputs for the first three intervals are identical. This is expected, because the agent has little information with which to infer the system behaviour at the beginning of the experiment. After this initial stage, the experiments diverge. The differences in the experiments are relatively minor, suggesting that there is a 'core' experimental design which works well over the distribution; the agent introduces slight deviates to maximise its effectiveness for different regions of parameter space.

## Discussion

In this work we demonstrated a novel application of reinforcement learning to the optimal design of biological experiments. The problem was formulated as the maximisation of the determinant of the Fisher Information matrix (D-optimal design) and we build on previous work that has demonstrated the applicability of Fisher information based methods to non-linear biological systems [12–18]. First the efficacy of the approach was tested using the FQ-learning algorithm to design optimal experiments in the unrealistic setting in which the method has prior knowledge of the parameter values we seek to identify. Positive results here indicated that this approach had the potential to design optimal experiments. We then introduced algorithm refinements that focussed on eliminating the dependence on the prior knowledge of parameter values. The dependence on the true parameter values is a limitation of other OED works [14–18], which require *ad hoc* verification [14] or other workarounds [16–18]. To decouple the RL controller from the true parameters, we used a recurrent neural network to make decisions based on a full experimental history of past measurements and experimental inputs. We showed that this approach produces an agent that can effectively design optimal experiments on systems sampled from a distribution over parameters, performing similarly to an MPC with explicit knowledge of the parameter values.

OED work has been undertaken using local sensitivity based methods via the Fisher information, methods based on global sensitivity analysis (GSA), and Bayesian methods. Fisher information based techniques have been limited to local analysis around a nominal parameter guess. Like the method developed in this paper GSA and Bayesian approaches allow global optimisation over a range of parameter values, but can be computationally expensive. OED methods based on GSA have shown reduced parameter uncertainty compared to local methods for Michaelis-Menton enzyme kinetics [41], Lithium-ion batteries [42] and chemical synthesis [43]. Bayesian approaches have been shown effective in systems biology applications [7–9]. Thorough comparisons of these approaches and the novel approach developed here are key directions for future work. However, GSA and Bayesian approaches often rely on evaluating expectations using methods such as Monte Carlo simulation [7–9] or the point estimate method [42, 43], which, for online experimental design, need to be determined during the experiment. This can be computationally expensive, potentially limiting the speed at which experiments can be done. Previous Bayesian work has been limited to greedily optimising over the next time step only [7] or choosing from a limited set of experimental designs [9, 11] and recent work has focused on alleviating these issues [44]. Here, we confirmed that training an RL controller over samples from a parameter distribution can result in an encoding of prior parameter knowledge and therefore optimisation globally with respect to the Fisher information. Because querying the RL controller for experimental inputs simply means executing the forward pass of a neural network, experimental decisions can be made in fractions of a second

(see Methods) which will enable real time, online OED. However, the price for this is a potentially lengthy training process which must be completed prior to the experiment. The longest training times in this work for our RT3D algorithm were around 12 hours of simulation time on a computing cluster or around 15 hours on a personal laptop. Reinforcement learning has been shown to scale to large problems in both discrete and continuous learning spaces. However, these applications can require training times and hardware that could be prohibitive for use in a biology laboratory setting. We have shown that this approach is feasible for a simple bacterial growth model and a realistic model of gene transcription (Figs E and F in S1 Text). A key direction for future work will be to assess and optimise the scalability of training our method on more complex models and for more complex design tasks, such as simultaneous model discrimination and parameter inference. We demonstrated the capability to learn over a uniform parameter distribution. Future work could be done to evaluate performance when trained over more complex distributions that are, e.g. non-symmetric or multi-modal. Because the method developed here learns by sampling the distribution, these alternatives would be straightforward to implement. Indeed, the flexibility to learn over any distribution presents a compelling reason to use reinforcement learning for OED.

Here, we have focussed on D-optimal design by maximising the determinant of the FIM. D-optimal design has a number of desirable qualities, making it the most common FIM based design metric [41]. However, other metrics, such as maximising the trace of the FIM (A-optimality) or maximising the minimum eigenvalue of the FIM (E-optimality) could be used. There are trade-offs for using different FIM based metrics [45], which has lead to investigations of modified and multicriteria objective functions. For instance, the combination of D and E-optimality [46], incorporation of a measure of curvature to improve experimental designs for non-linear dynamic models [47], summing the information matrices across multiple experiments run in parallel [48] and parameter correlation reduction [49, 50]. Furthermore, there are Bayesian objectives such as minimising the entropy of the posterior parameter distribution [9, 11]. Each of these alternative objectives could be incorporated into the reinforcement framework—a compelling direction for future work.

Overall, we have developed and demonstrated the potential for reinforcement learning to be used for OED. As the systems we build and characterise in biology continue to increase in complexity, automated experimental design tools will become ever more important. Furthermore the generality of this approach means it can be applied in many areas of science and engineering.

## Materials and methods

### Formulation of the optimal experimental design problem

Optimal experiments will be designed on systems which can be described by a set of non-linear differential equations:

$$\frac{d\mathbf{X}}{dt} = \mathbf{F}(\mathbf{X}, \boldsymbol{\theta}, \mathbf{u}), \tag{1}$$

where $\mathbf{X}$ is a vector of state variables, $\boldsymbol{\theta}$ is a vector of parameters and $\mathbf{u}$ is a vector of experimental inputs. System measurements are assumed of the form $\mathbf{Y} = \mathbf{X}^{\mathrm{m}} + \boldsymbol{\epsilon}$, where $\mathbf{X}^{\mathrm{m}}$ are the measurable state variables and $\boldsymbol{\epsilon}$ is a Gaussian noise term. We consider optimal experimental design tasks with the goal of inferring accurate estimates of the system parameters, $\boldsymbol{\theta}$. We define our objective as the determinant of the Fisher Information matrix, $|I|$. This is called a D-optimality criterion. The theory demonstrating that D-optimality corresponds to maximally accurate parameter inferences holds only for linear systems with Gaussian errors, but previous

work has shown that this same criterion can be successfully applied to non-linear systems [12–15]. We follow an established method [14, 15] to obtain $I$ from the system equations (Eq 1). First we obtain time derivatives for the sensitivity of each of the state variables with respect to each of the parameters:

$$\frac{d}{dt}\frac{\partial \mathbf{X}^{\mathrm{m}}}{\partial \theta_j} = \frac{\partial \mathbf{F}}{\partial \theta_j} + \frac{\partial \mathbf{F}}{\partial \mathbf{X}^{\mathrm{m}}}\frac{\partial \mathbf{X}^{\mathrm{m}}}{\partial \theta_j}.$$

(2)

The scale of parameters can vary, which can lead to poor conditioning of the Fisher information matrix [31]. To remedy this, the sensitivities are scaled by the parameter values, which is equivalent to using logarithmic sensitivities. The scaled sensitivities are

$$\bar{\mathbf{X}}^{\mathrm{m}}_{\theta_j} = \frac{\partial \mathbf{X}^{\mathrm{m}}}{\partial \log(\theta_j)} = \theta_j \frac{\partial \mathbf{X}^{\mathrm{m}}}{\partial \theta_j}.$$

Writing Eq 2 in terms of the scaled sensitivities yields

$$\frac{d\bar{\mathbf{X}}^{\mathrm{m}}_{\theta_j}}{dt} = \theta_j \frac{\partial \mathbf{F}}{\partial \theta_j} + \frac{\partial \mathbf{F}}{\partial \mathbf{X}^{\mathrm{m}}}\bar{\mathbf{X}}^{\mathrm{m}}_{\theta_j}.$$

We assume that measurement error $\boldsymbol{\epsilon}$ is normally distributed with variance equal to 5% of the measured quantity: $\boldsymbol{\epsilon} = \mathcal{N}(0, \boldsymbol{\sigma}^2), \boldsymbol{\sigma}^2 = 0.05\mathbf{X}^{\mathrm{m}}$, and we assume measurements are independent. The time derivative of the scaled FIM can be written as [33]

$$\frac{d}{dt}I_{jk}(\mathbf{u}, \boldsymbol{\theta}, t) = \bar{\mathbf{X}}^{\mathrm{m}}_{\theta_j}\Sigma^{-1}(t)\bar{\mathbf{X}}^{\mathrm{m}}_{\theta_k},$$

where

$$\Sigma(t) = \begin{bmatrix} \sigma_1^2 X_1^{\mathrm{m}} & & \\ & \ddots & \\ & & \sigma_n^2 X_n^{\mathrm{m}} \end{bmatrix}$$

is a diagonal matrix, where $X_i^{\mathrm{m}}$ and $\sigma_i^2$ are the $i$-th measurable state variable and the associated variance, respectively. The FIM can then be determined by integration over an experiment, assuming that $I(t = 0) = 0$

$$I_{jk}(\mathbf{u}, \boldsymbol{\theta}, 0, t_f) = \int_0^{t_f} \bar{\mathbf{X}}^{\mathrm{m}}_{\theta_j}\Sigma^{-1}(t)\bar{\mathbf{X}}^{\mathrm{m}}_{\theta_k}dt.$$

For the optimisation objective we use the determinant of the FIM (the D optimality score). Because this can vary over orders of magnitude, the optimality criterion is taken to be the logarithm of the determinant of the FIM:

$$\Theta_D(\mathbf{u}, \boldsymbol{\theta}, 0, t_f) = \log(|I(\mathbf{u}, \boldsymbol{\theta}, 0, t_f)|).$$

(3)

We use the shorthand $I_\tau = I(\mathbf{u}, \boldsymbol{\theta}, 0, t_f)$, where $t_f$ is the time at the beginning of time step $\tau$. The Python API of the CasADi library was used to solve all differential equations in our analysis [51].

## Neural fitted Q-learning algorithm

In neural FQ-learning [28], a value function is learned that maps observation-action pairs to values, $Q(o, a)$. This value function is represented by a neural network. We define a transition

as the tuple $(o, a, r, o', d)$ specifying, respectively, an observation, an action, a reward, a subsequent observation, and an indicator that is 1 if the episode terminated during this transition and otherwise is 0. Here $a$ is the action taken after observing $o$, while $r$ and $o'$ are the reward and the observation that result from taking action $a$. The state can be continuous, but the action is limited to a discrete set of feasible values. From a sequence of these state transitions, a sequence of Q-learning targets is generated as:

$$Q(o, a)_{target} = r + (1 - d)\gamma \max_a Q(o', a) \tag{4}$$

Here the term $\max_a Q(o', a)$, where $a$ is an action that can be taken by the agent, gives an estimate of the total future reward obtained after observing $o'$. This is weighted by $\gamma$, the discount factor, which dictates how heavily the possible future rewards affect decisions. The neural network is trained on the set of inputs and targets generated from all training data seen so far (Algorithm 1). A training iteration takes place after each episode, resulting in Algorithm 2. We used the Adam optimiser [52] to train the neural network because of its ability to dynamically adapt the learning rate, which is favourable when implementing reinforcement learning with a neural network [53].

In this work we use an $\epsilon$-greedy policy in which a random action is chosen with probability $\epsilon$ and the action $a = \max_a Q(o, a)$ is chosen with probability $1 - \epsilon$. The explore rate was initially set to $\epsilon = 1$ and decayed as $\epsilon = \text{clip}(\log_{10}(\frac{e}{A}), 0, 1)$ where $e$ is the episode number, starting at 0, and $A$ is a constant that dictates the rate of decay. The clip function checks if its first argument is within the interval defined by the second and third arguments and, if it is not, clips it to the nearest edge of the interval. In this work, $A$ was set to the total number of episodes used in a given training simulation divided by 11. This choice ensures that the explore rate reaches 0 before the end of training. This $\epsilon$-greedy strategy is a widely used policy that has been proven effective [19, 20] and is easy to implement. The inputs to the neural network are continuous. The input layer is of size 11 if the elements of the FIM are included, otherwise it is of size 2 to account for the single measured variable and the time step index. The neural network contains two hidden layers with 100 neurons each, all of which use the ReLU activation function. The linear output layer consists of 100 neurons, accounting for the discretisation of the two dimensional action space into 10 bins along each axis. When a recurrent network is used two additional GRU layers were added each with 64 neurons to process the sequence of previous observations and actions, the output of which is concatenated to the current observation before being passed to the rest of the network. In this case, the network is structured similarly to the value network in Fig 3A.

**Algorithm 1** FQ-iteration

```
1: input: D            ▷ memory of transitions encountered so far
2: reinitialise Q network
3: inputs = {o  ∀ o ∈ D}
4: targets = {r + γ(1 − d)max_a Q_i(o')  ∀ a, r, o', d ∈ D}
5: train Q network on (inputs, targets) → Q
6: return Q
```

**Algorithm 2** FQ-learning

```
1: hyperparameter: E        ▷ number of episodes
2: hyperparameter: T        ▷ number of time steps in each episode
3: for e in 1 to E do
4:   o = env.reset()        ▷ reset env and get initial o
5:   for τ in 0 to T do
6:     a = π(o, Q_N)         ▷ get action based on current policy
7:     (r, o', d) = env.step(a)      ▷ interact with env and observe
                                        transition
```

## 1) Define distribution over parameters

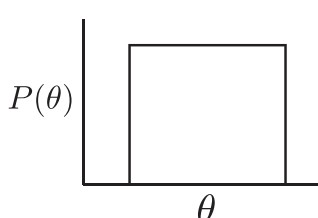

## 2) Generate data using random policy

For every episode take a new sample $\theta \sim P(\theta)$

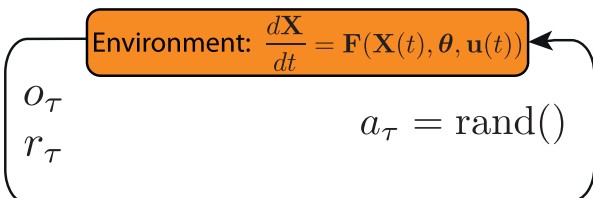

## 3) Learn the value function

Sequence of observations, actions and returns

$(o, a, G)_0$

$\vdots$

$(o, a, G)_N$

Fit value function → Agent

**Fig 5. Testing agents that do not have prior access to simulation parameter values.** 1) A distribution of parameters was chosen (uniform distribution in this case). 2) Data was generated on simulated chemostat models using a random policy for action selection. 3) Agents were trained to predict the observed returns.

```
8:     𝒟 ← 𝒟 + (o, a, r, o', d)        ▷ add transition to memory
9:     o = o'        ▷ update current observation
10:    Q = FQ_iteration(𝒟)        ▷ update agent's policy
11: return Q
```

## Value fitting

The process used for evaluating an agent's ability to learn a value function is illustrated in Fig 5. First, a training set of 10,000 experiments was generated by sampling 10,000 different parametrisations of the chemostat model from a uniform distribution over parameters, with maximal and minimal values as in Table 1. For each parametrisation, 10 random experimental inputs were applied, sampled uniformly from the discrete set of actions. This procedure was repeated to generate an independent testing set. For each experiment the return was calculated as the sum of the rewards obtained after visiting each observation-action pair through the experiment, $G(o_\tau, a_\tau) = \sum_{i=\tau}^{10} r_i$. This results in two independent datasets consisting of $N = 10,000 \times 10$ data-points, where each data-point is composed of an observation, $o$, an action, $a$, and the corresponding return, $G$. The value functions of each agent was fitted to the training data set using repeated FQ-iterations (Algorithm 1), where the Q-learning targets were the returns, $G$. (Note, this training scenario is distinct from the episode-based reinforcement learning strategy.) This procedure tests an agent's ability to learn the value function from a random policy applied to a range of model parameterisations. Because the parameter values are different for each experiment, the agent must learn to infer where it is in parameter space to optimally predict the value of a given observation-action pair.

## Twin delayed deep deterministic policy gradient

Twin delayed deep deterministic policy gradient (T3D) [30] is an off-policy algorithm for continuous deep reinforcement learning (Algorithm 3). It is based on a previously established algorithm called deep deterministic policy gradient (DDPG) [54], but introduces a few modifications to improve learning stability. The DDPG algorithm is closely related to Q-learning and can be thought of as Q-learning adapted for continuous action spaces. Like deep Q-learning, DDPG uses a neural network to approximate and learn a Q-function $Q(o, a)$ which maps observation action pairs to a value. In addition, DDPG also learns a policy, $a = \pi(o)$, which is represented by a second neural network. The policy network maps states to actions and is trained to choose the action, $a$, that maximises the value of the state action pair for the given observation, $o$, according to the value network: $a = \mathrm{argmax}_a\, Q(o, a) \approx Q(o, \pi(o))$.

The DDPG algorithm proceeds as follows. As in FQ-learning, a transition is defined as the tuple $(o, a, r, o', d)$ specifying, respectively, an observation, action, reward, a subsequent observation and an indicator that is 1 if the episode terminated during this transition and otherwise is 0. As the agent learns, it stores observed state transitions in a replay buffer, $\mathcal{D}$, which can be though of as its memory. Two tricks are used to increase stability of the learning process in DDPG. Firstly, at each update a random sample, $\mathcal{B}$, of past experience is taken from the replay buffer to reduce the temporal correlation in the updates. Secondly, target networks $Q_{targ}$, $\pi_{targ}$ are used to generate the Q-learning targets. The parameters of these networks update slowly to the parameters of $Q$ and $\pi$ by Polyak averaging, $\theta_{targ} = \rho\theta_{targ} + (1 - \rho)\theta$. This reduces the dependence of the target on the trained parameters and further increases stability.

Three further additions to DDPG lead to the T3D algorithm. First, the policy network updates are delayed by updating half as frequently as the Q-network. Second, to address a common failure mode of DDPG in which the policy can exploit incorrect sharp peaks in the Q-function, the target policy is smoothed by adding random noise to the target actions which effectively regularises the algorithm:

$$a'(o') = \text{clip}(\pi_{targ}(o') + \text{clip}(\xi, -c, c), a_{low}, a_{high}), \ \ \xi \sim \mathcal{N}(0, \sigma)$$

where $c$ is an upper bound on the absolute value of the noise, $a_{low}$ and $a_{high}$ are lower and upper bounds on the target action respectively, and $\sigma$ is the standard deviation of the noise. Finally, because all Q-learning methods involve maximising over target actions, they are prone to overestimate the Q-function. To reduce this tendency, double Q-learning is implemented in T3D: two Q-functions, $Q_1$ and $Q_2$, are learned and the one that gives the smaller value is used to calculate the Q-learning target. From a sequence of state transitions, $\mathcal{B}$, sampled from the replay buffer a sequence of Q-learning targets, $y$, is created according to:

$$y(r, o', d) = r + \gamma(1 - d) \min_{i=1,2} Q_{i,targ}(o', a') \ \ \forall \ \ (o, a, r, o', d) \in \mathcal{B}$$

The term $(1 - d) \min_{i=1,2} Q_{i,targ}(o', a')$ gives an estimate of the total future reward obtained after entering state $o'$. The networks $Q_1$ and $Q_2$ are trained on the set of inputs by regressing to the targets with the following losses.

$$L_1(\mathcal{D}) = \mathbb{E}_{\mathcal{B}}[(Q_1(o, a) - y(r, o', d)^2]$$

$$L_2(\mathcal{D}) = \mathbb{E}_{\mathcal{B}}[(Q_2(o, a) - y(r, o', d)^2]$$

Then, every other update, the policy network is updated by training it to maximise $Q_1$:

$$\max_{\theta} \mathbb{E}_{\mathcal{B}}[Q_1(o, \pi_\theta(o))] \tag{5}$$

Finally, the target networks are updated:

$$\theta_{Q_{1,targ}} = \rho\theta_{Q_{1,targ}} + (1 - \rho)\theta_{Q_1}$$

$$\theta_{Q_{2,targ}} = \rho\theta_{Q_{2,targ}} + (1 - \rho)\theta_{Q_2}$$

$$\theta_{\pi_{targ}} = \rho\theta_{\pi_{targ}} + (1 - \rho)\theta_\pi$$

We use an $\epsilon$-greedy policy adapted to the continuous action space; a random action is uniformly chosen between $a_{low}$ and $a_{high}$ with probability $\epsilon$ and the action $a = \text{clip}(\pi(s) + \mathcal{N}(0, 0.2\epsilon), a_{low}, a_{high})$ is chosen with probability $1 - \epsilon$. The explore rate $\epsilon$ was set to decay exponentially as training progressed. The explore rate was initially set to $\epsilon = 1$ and decayed as

$\epsilon = \text{clip}(\log_{10}(\frac{e}{A}), 0, 1)$ where $e$ is the episode number and $A$ is a constant that dictates the rate of decay, where $A$ is equal to the number of episodes divided by eleven. This ensures the explore rate reaches 0 before the end of training. The Adam optimiser [52] was used to train the neural networks, because of its ability to dynamically adapt the learning rate, which is favourable when implementing reinforcement learning with a neural network [53].

**Algorithm 3** T3D

```
 1: hyperparameter: E          ▷ number of episodes
 2: hyperparameter: T          ▷ number of time steps in each episode
 3: for e in 1 to E do
 4:   o = env.reset()          ▷ reset env and get initial o
 5:   for τ in 0 to T do
 6:     a = π(o)               ▷ get action based on current policy
 7:     (r, o′, d) = env.step(a)        ▷ interact with env and observe
                                           transition
 8:     𝒟 ← 𝒟 + (o, a, r, o′, d)         ▷ add transition to memory
 9:     update_count = 0
10:     if t%update_frequency = 0 then
11:       ℬ ∼ 𝒟
12:       a′ = clip(π_targ(o′) + clip(ξ, −c, c), a_low, a_high),   ξ ∼ 𝒩(0, σ)  ∀  o ∈ ℬ
13:       y(r, o′, d) = r + (1 − d)γ min_{i=1,2} Q_{i,targ}(o′, a′)  ∀  (o, a, r, o′, d) ∈ ℬ
14:       Train Q_1, Q_2 networks on y(r, o′, d)
15:       if update_count % policy_delay == 0 then
16:         Train π network to max_θ 𝔼_ℬ[Q_1(o, π_θ(o))]
17:         θ_targ = ρθ_targ + (1 − ρ)θ         ▷ update target networks
18:     o = o′        ▷ update current observation
19: return π        ▷ return trained policy
```

The details of our neural networks are as follows (see Fig 3A). The value network contains a GRU cell composed of two layers each with 64 neurons. This takes as input all system measurements, $\mathbf{Y}$, and experimental inputs, $\mathbf{u}$, seen so far in the current experiment. The GRU cell is a type of recurrent layer which excels at processing sequences such as time series data. The output of the GRU is concatenated with the current system measurements, $\mathbf{Y}_\tau$, the current time step, $t$, and the current action, $\mathbf{u}_\tau$. These are fed into a feed forward network composed of two hidden layers, each with 128 neurons. The output of the value network is an estimate of the value of the supplied experimental input, $\mathbf{u}_\tau$, given the sequence of observables. The policy network contains a recurrent GRU cell composed of two layers each with 64 neurons. This takes as input all measurements and experimental inputs seen so far in the current experiment. The output of the GRU is concatenated with the current observation, composed of the current system measurements and the current time step, and fed into a feed forward network composed of two hidden layers, each with 128 neurons. The output of the policy network is the experimental input that is estimated to have the highest reward value for the given sequence of observations. TensorFlow version 2.4.1 [55] was used to implement all neural networks.

Because training a RL controller takes place over a large number of independent episodes, it is possible to gather experience from multiple episodes in parallel. We took advantage of this by running experimental simulations in parallel using the functionality provided by the CasADi library. This means that all of the computationally demanding aspects of the training process are parallelised. The training of the neural networks using TensorFlow is implemented in parallel and can be run on a GPU or a multicore CPU. The simulation of the experimental system using CasADi is parallelised and can take advantage of a multicore CPU. Experimental simulations were run in batches of 10 parallel simulations, but this number could be increased to take advantage of more computing resources. The average training time for the RT3D algorithm training on the chemostat system over a parameter distribution was 11.73 hours with

standard deviation 0.91 hours over 20 total training runs on a computing cluster with 40 CPU cores and a GeForce GTX 1080 Ti. To assess the training time on a smaller scale personal computer, a single RT3D agent was trained on a 13-inch MacBook pro with a 2GHz Quad core Intel i5 processor with no GPU. The training time was 15.48hrs. Online experimental inputs can be determined in parallel with little to no additional time cost. For instance, to determine 100 parallel experiments on the chemostat system takes approximately 0.005s on a desktop computer with a modest GPU (NVIDIA Quadro P2000) and 0.02 seconds on a MacBook Pro with a 2GHz Quad-Core Intel Core i5 and no GPU. (Note that this time is dependent only on the size of the neural network used and so is not directly a function of system complexity, although more complex systems may require the use of larger neural networks.)

## Scaling neural network inputs and outputs

To prevent network instability and improve convergence, inputs to neural networks are often scaled [56]. The inputs to the neural networks for all reinforcement learning agents were scaled to be approximately between 0 and 1. This was done by dividing each input by a scalar normalisation constant. At the nominal parameter values, the carrying capacity of the chemostat system is $48 \times 10^9$ and so using a normalisation coefficient of $50 \times 10^9$ ensures that all population measurements are between 0 and 1 before entering the network. This normalisation of the population measurements was used in the first FQ-learning results section. The time step, $\tau$ is known to be an index between 0 and 10 and so a normalisation constant of 10 was used for all sections. The elements of the FIM comprising the agent's observations present more of a challenge. To find a suitable normalisation constant for each of these, the highest value of each FIM element during the agent's exploration phase was found during a trial training run, where the portion of training in which the explore rate is equal to 0 was simulated. This value was used as the normalisation constant.

When sampling from a parameter distribution the normalisation can be more challenging. Some regions of parameter space led to instability in the simulation output. Such simulations were discarded. To limit the number of episodes that were discarded, a square root was applied to the population measurement before being normalised by $1 \times 10^5$, any episodes with observations greater than 1 after normalisation were taken to be unstable and discarded. Approximately 2.5% of training episodes were discarded. The square root increases the allowable population range while still allowing sufficient separation between different population measurements for the agent to learn. (We attempted application of a $\log_{10}$ scaling to the population measurements but found that led to poor performance. We hypothesise that because the majority of episodes are stable and have populations that vary within a single of magnitude, the agent was unable to sufficiently distinguish between different population measurements under that logarithmic scaling.) For consistency, the output of the neural networks was also scaled to be between 0 and 1 by dividing the Q-learning targets by 100.

## Model predictive control

A model predictive controller (MPC) [57] uses a calibrated model of a system (Eq 1) to predict optimal control inputs. Assuming that the parameter values are known, the MPC integrates the model over a predefined time interval to optimise an objective function with respect to the control inputs **u**. In this work, the objective function was the optimality score of an experiment (Eq 3). The time horizon the MPC optimises over is a hyperparameter to be chosen. We consider two variants of MPC. The first optimizes over a single time interval; we refer to this as the

one step ahead optimiser (OSAO). At time step $\tau$ the OSAO solves the optimisation problem

$$\mathbf{u}_\tau = \arg \max_{\mathbf{u}} (\log|I_{\tau+1}| - \log|I_\tau|). \tag{6}$$

(Note, the term $\log|I_\tau|$ has already been determined by the start of this interval and so does not contribute to the optimisation task; we include it here to simplify the interpretation of the full MPC optimality criterion below.) The second variant is a controller that optimises over the full timeseries of $N$ intervals simultaneously; we refer to this as the MPC. At the beginning of the experiment the MPC uses the model to optimise over the full experiment, choosing each input that will be applied by solving the optimisation problem:

$$[\mathbf{u}_0, \mathbf{u}_1, \ldots, \mathbf{u}_{\mathcal{T}-1}] = \arg \max_{[\mathbf{u}_0, \mathbf{u}_1, \ldots, \mathbf{u}_{\mathcal{T}-1}]} \log|I_{\mathcal{T}}| \tag{7}$$

where $\mathcal{T}$ is the number of time steps in the experiment. To solve the optimisation problems for both the OSAO and MPC the non-linear solver IPOPT [58] was called from the CasADi library [51].

## Parameter fitting

To confirm that our optimality scores correspond to improved parameter estimation accuracy, we used simulated experiments to assess the correlation between high D-optimality scores and low covariance in the resulting parameter estimates. To generate simulated data, we simulated the model using the given experimental design and added normally distributed observation error with a variance equal to 5% of the corresponding output. Thirty independent simulated datasets were generated. When assuming prior knowledge of the parameter values the model was simulated using the nominal parameter values. When a parameter distribution was used, parameters were sampled from the distribution before each simulated experiment. The non-linear solver IPOPT [58] was called from the CasADi library [51] to infer parameter estimates from the simulated data.

## Supporting information

**S1 Text. Supporting information.**
(PDF)

## Author Contributions

**Conceptualization:** Neythen J. Treloar, Nathan Braniff, Brian Ingalls, Chris P. Barnes.

**Funding acquisition:** Brian Ingalls, Chris P. Barnes.

**Investigation:** Neythen J. Treloar.

**Methodology:** Neythen J. Treloar, Nathan Braniff, Brian Ingalls, Chris P. Barnes.

**Software:** Neythen J. Treloar.

**Supervision:** Brian Ingalls, Chris P. Barnes.

**Visualization:** Neythen J. Treloar.

**Writing – original draft:** Neythen J. Treloar, Nathan Braniff, Brian Ingalls, Chris P. Barnes.

**Writing – review & editing:** Neythen J. Treloar, Nathan Braniff, Brian Ingalls, Chris P. Barnes.

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
