## [Decision Letter · Decision Letter 0]

3 Aug 2022

Dear Dr Treloar,

Thank you very much for submitting your manuscript "Deep Reinforcement Learning for Optimal Experimental Design in Biology" for consideration at PLOS Computational Biology.

As with all papers reviewed by the journal, your manuscript was reviewed by members of the editorial board and by several independent reviewers. In light of the reviews (below this email), we would like to invite the resubmission of a significantly-revised version that takes into account the reviewers' comments.

We cannot make any decision about publication until we have seen the revised manuscript and your response to the reviewers' comments. Your revised manuscript is also likely to be sent to reviewers for further evaluation.

Sincerely,

Vassily Hatzimanikatis

Associate Editor

PLOS Computational Biology

Mark Alber

Deputy Editor

PLOS Computational Biology

Reviewer's Responses to Questions

**Comments to the Authors:**

Reviewer #1: This manuscript presents a deep reinforcement learning approach for the design of optimal experiments in biological systems.

Although this is a nice contribution, it needs to be improved in several ways. The main issues are:

1- the literature review in the introduction should be updated, especially regarding:

(i) links between OED and control theory (see e.g. surveys by Gevers, or Hjalmarsson; also papers from Pronzato such as https://doi.org/10.1016/j.automatica.2007.05.016 )

(ii) other recent related approaches, e.g. https://arxiv.org/abs/2202.00821

2- previous works (e.g. by Schenkendorf and others) have illustrated the use of global sensitivity analysis in OED to address some of the concerns outlined in this study. Such works should be cited/discussed.

3- the authors have focused on D-optimal design criterion. Previous works (also not cited in this study) have shown a trade-off between different FIM-based metrics (e.g. D and mod-E). Maybe a multicriteria approach could be used (as previously done by several groups). It would be nice if the authors discuss this aspect.

4- I think the author should also check (and cite/discuss) the excellent survey by Macchietto ( https://doi.org/10.1016/j.ces.2007.11.034 ) which, although not very recent, discusses in detail several of the above issues and the related literature

5- the authors show that a deep reinforcement learning approach performs favourably in comparison with a one-step ahead optimisation algorithm and a model predictive controller using a rather simple case study (bacterial growth in a chemostat). How scalable is their approach for more realistic applications?

6- also regarding scalability, recent alternative BO-based approaches should also be discussed (e.g. https://doi.org/10.23919/ACC45564.2020.9147824 )

Minor issues:

- some refs are incomplete

Reviewer #2: The manuscript by Treloar and colleagues presents a novel methodology to optimally design maximally informative experiments for the task of parameter inference for mathematical models describing biological systems. The method uses reinforcement learning to predict the informative content of potential experiments: while it does use a well-established technique (FIM-based metric) to estimate information content, the approach documented by the authors addresses a key problem in current approaches to OED, that is the need to either start from a neighbourhood of the actual value of the parameters when performing model calibration, or "paying the cost" of expensive Bayesian methods.

Overall the approach presented is sound, the paper is well structured and presented. I only have a few comments, offered, in order of importance, below:

- Given the audience of this journal, I believe that the manuscript would greatly benefit from a section that outlines the key elements of Reinforcement Learning and clarifies some of the jargon (Fitted Q, value function etc);

- While the authors compare their RL agents to MPC and one step ahead optimization I think that it would be helpful to show how the solutions the best agent identifies compares to other optimisation schemes commonly used in dynamic experimental design (e.g. eSS). Also, connected, RL is presented as a solution but the rationale for using ML in the first place is not entirely clear.

- In Section 4.4 I found the speculation on the reason for the underperformance ("problem too complex") requiring a level of suspension of judgement that I was not entirely comfortable with. Could the authors simplify the problem (less control intervals or less discrete values of the inputs) to show that "problem complexity" is indeed the issue?

- At page 5, where the chemotast model is introduced, the authors should review the vay the parameters are introduced as it seems inconsistent with the text that explains them;

- The manuscript is sprinkled with a parentheticals, some dispensable (e.g. I don't think this audience needs to be told what an auxotroph is). I would encourage the authors to go over the text and ask, for each sentence between parentheses, whether it is necessary. If so, make a proper sentence of it, otherwise eliminate;

- At page 11: "the determinant of the Fisher Information" should read "the determinant of the Fisher Information matrix"

Overall I genuinely believe this is an excellent piece of work and the authors should be commended for their contribution.

**Have the authors made all data and (if applicable) computational code underlying the findings in their manuscript fully available?**

Reviewer #1: Yes

Reviewer #2: Yes

PLOS authors have the option to publish the peer review history of their article (what does this mean?). If published, this will include your full peer review and any attached files.

Reviewer #1: No

Reviewer #2: No
---

## [Decision Letter · Decision Letter 1]

22 Oct 2022

Dear Dr Treloar,

Thank you very much for submitting your manuscript "Deep Reinforcement Learning for Optimal Experimental Design in Biology" for consideration at PLOS Computational Biology. As with all papers reviewed by the journal, your manuscript was reviewed by members of the editorial board and by several independent reviewers. The reviewers appreciated the attention to an important topic. Based on the reviews, we are likely to accept this manuscript for publication, providing that you modify the manuscript according to the review recommendations.

Sincerely,

Mark Alber, Ph.D.

Section Editor

PLOS Computational Biology

Mark Alber

Section Editor

PLOS Computational Biology

Reviewer's Responses to Questions

**Comments to the Authors:**

Reviewer #1: I find the revised version satisfactory. There are a couple of minor things to be fixed:

- ref 10 must be corrected (it shows the funding info, not the journal)

- the annotated colored version is badly compiled regarding refs etc., but I was able to see the changes anyway

**Have the authors made all data and (if applicable) computational code underlying the findings in their manuscript fully available?**

Reviewer #1: Yes

PLOS authors have the option to publish the peer review history of their article (what does this mean?). If published, this will include your full peer review and any attached files.

Reviewer #1: No

Figure Files:

Data Requirements:

Reproducibility:

References:

---

## [Editor Report · Decision Letter 2]

31 Oct 2022

Dear Dr Treloar,

We are pleased to inform you that your manuscript 'Deep Reinforcement Learning for Optimal Experimental Design in Biology' has been provisionally accepted for publication in PLOS Computational Biology.

Best regards,

Mark Alber, Ph.D.

Section Editor

PLOS Computational Biology

Mark Alber

Section Editor

PLOS Computational Biology

---

## [Editor Report · Acceptance letter]

15 Nov 2022

PCOMPBIOL-D-22-00713R2 

Deep Reinforcement Learning for Optimal Experimental Design in Biology

Dear Dr Treloar,

I am pleased to inform you that your manuscript has been formally accepted for publication in PLOS Computational Biology. Your manuscript is now with our production department and you will be notified of the publication date in due course.

With kind regards,

Anita Estes
